# Differences in diacylglycerol acyltransferases expression patterns and regulation cause distinct hepatic triglyceride deposition in fish
Xiaojun Xiang[1], Renlei Ji[1], Shangzhe Han[1], Xiang Xu[1], Si Zhu[1], Yongnan Li[1], Jianlong Du[1], Kangsen Mai[1,2] & Qinghui Ai [1,2]✉

Triglyceride (TAG) deposition in the liver is associated with metabolic disorders. In lower vertebrate, the propensity to accumulate hepatic TAG varies widely among fish species. Diacylglycerol acyltransferases (DGAT1 and DGAT2) are major enzymes for TAG synthesis. Here we show that large yellow croaker (*Larimichthys crocea*) has significantly higher hepatic TAG level than that in rainbow trout (*Oncorhynchus mykiss*) fed with same diet. Hepatic expression of *DGATs* genes in croaker is markedly higher compared with trout under physiological condition. Meanwhile, DGAT1 and DGAT2 in both croaker and trout are required for TAG synthesis and lipid droplet formation in vitro. Furthermore, oleic acid treatment increases *DGAT1* expression in croaker hepatocytes rather than in trout and has no significant difference in *DGAT2* expression in two fish species. Finally, effects of various transcription factors on croaker and trout *DGAT1* promoter are studied. We find that DGAT1 is a target gene of the transcription factor CREBH in croaker rather than in trout. Overall, hepatic expression and transcriptional regulation of DGATs display significant species differences between croaker and trout with distinct hepatic triglyceride deposition, which bring new perspectives on the use of fish models for studying hepatic TAG deposition.

Liver plays an essential role in maintaining metabolic homeostasis and aberrant accumulation of hepatic triglyceride (TAG) is an important factor in the pathogenesis of metabolic diseases[1,2]. Extensive studies showed that the propensity to accumulate hepatic TAG vary markedly among ethnicities due to the differences in genetic background and environmental factors in mammals[3–5]. In the lower vertebrate, fish liver is the central organ responsible for lipid metabolism and the propensity of TAG to accumulate in the liver differs substantially among fish species. Several fish species stored excess lipid in viscera adipose tissue preferentially, such as Nile tilapia (*Oreochromis niloticus*), while Atlantic salmon (*Salmo salar*) and black seabream (*Acanthopagrus schlegelii*) would prefer to store lipid in muscle but not in liver[6–8]. Besides the adverse effect on growth performance and health, excessive hepatic TAG accumulation-induced inflammation in farmed fish causes great harm to the aquaculture industry and poses a serious threat to food safety[9,10]. However, the molecular mechanisms of hepatic TAG deposition disparity in fish remains poorly understood.

Variation in genes involved in hepatic lipid metabolism is associated with the difference in hepatic fat content in mammals[11,12]. Diacylglycerol acyltransferases (DGAT1 and DGAT2) are the rate-limiting enzymes in TAG synthesis and lipid droplet formation in most eukaryotes[13–16]. DGATs are also regarded as candidate genes contributing to genetic control of depot-specific TAG deposition in mammals[17,18]. Liver-specific over-expression of DGAT2 in mice increased hepatic TAG level, and the increase of DGAT1 expression has been observed in individuals with non-alcoholic fatty liver[19,20]. DGATs in fish share high sequence similarity with DGATs in mammals[21,22]. However, whether DGATs are responsible for TAG synthesis in fish and display marked differences among fish species with distinct hepatic triglyceride deposition remains unclear.

[1]Key Laboratory of Aquaculture Nutrition and Feed (Ministry of Agriculture and Rural Affairs) & Key Laboratory of Mariculture (Ministry of Education), Ocean University of China, 5 Yushan Road, Qingdao, Shandong 266003, P.R. China. [2]Laboratory for Marine Fisheries Science and Food Production Processes, Qingdao National Laboratory for Marine Science and Technology, 1 Wenhai Road, Qingdao, Shandong 266237, People's Republic of China. ✉e-mail: qhai@ouc.edu.cn

Nutrition overload such as high fat diet (HFD) is the major cause of excessive hepatic TAG accumulation in both mammals and fish[23–26]. Previous studies showed that the expression of *DGAT1* and *DGAT2* was increased in the liver in response to HFD[27,28]. The specific role of DGAT1 and DGAT2 in HFD-induced fatty liver is divergent among different species. Inhibition of DGAT1 activity could alleviate HFD-induced hepatic TAG accumulation in mice, while reduced *DGAT2* expression, but not *DGAT1*, reduced hepatic TAG content in rats fed HFD[29,30]. Nonetheless, upon HFD feeding, the function and regulatory mechanism of DGATs in fish is still unknown. Previous studies have shown that many enzymes involved in TAG synthesis are regulated at the transcriptional level in response to diverse stimuli, especially nutrient signals[31,32]. Studies in mice indicated that liver-enriched transcription factor cAMP responsive element-binding protein (CREBH, encoded by *creb3l3*) plays an important role in hepatic TAG metabolism by regulating expression of target genes involved in lipid metabolism, including fatty acid synthase, apolipoprotein A-IV and *DGAT2*[33,34]. Meanwhile, CREBH is activated by nutritional conditions, such as fasting and HFD[35]. Furthermore, the stability and transcriptional activity of CREBH are modulated through several posttranslational modifications, such as phosphorylation, glycosylation and acetylation[36–38]. However, it remains unclear whether and how CREBH is involved in the regulation of DGATs expression in response to HFD.

Large yellow croaker and rainbow trout have unique features in vertebrate evolution and have become increasingly important for studying metabolic diseases[39–41]. Meanwhile, the storage depot for TAG is different between large yellow croaker and rainbow trout. Lipid deposition occurs mainly in the liver of large yellow croaker, whereas it is mostly found in the visceral adipose tissue of rainbow trout[42,43]. Moreover, HFD feeding induced excessive hepatic TAG accumulation in large yellow croaker rather than in rainbow trout[26,43,44]. Thus, large yellow croaker and rainbow trout are considered suitable subjects to investigate molecular mechanisms of hepatic TAG deposition disparity in fish.

In the present study, we explored the function of DGATs in hepatic TAG accumulation and discovered differences in DGATs between two fish species. This brings a perspective on the use of fish models for studying hepatic TAG deposition and improving the understanding of liver diseases. Identification of regulatory mechanisms contributing to hepatic TAG deposition disparity is of critical importance to develop novel therapies for HFD-induced fatty liver.

## Results

### Untargeted lipidomic analysis of croaker and trout livers
First, lipidomic analysis was performed in the liver of large yellow croaker and rainbow trout fed same nutritional diet to characterize the deposition of differential lipid species. The principal components analysis (PCA) showed that hepatic lipid profiles were apparently different between large yellow croaker and rainbow trout (Supplementary Fig. 1a). Hundreds of diverse lipids (455 species from 19 major lipid classes) were identified in the liver of large yellow croaker and rainbow trout, where TAG was the most enriched lipid class including 181 different species (Supplementary Fig. 1b). In addition, TAG showed significantly higher abundance in the liver of large yellow croaker than that in rainbow trout (Fig. 1a). A heat map suggested that relative contents of hepatic TAG and diacylglycerol (DAG) were higher in large yellow croaker than that in rainbow trout (Fig. 1b). Furthermore, large differences in specific DAG and TAG species were observed in large yellow croaker compared with rainbow trout livers (Fig. 1c and d).

### Bioinformatics analyses of DGAT1 and DGAT2 sequences
To gain insight into DGATs functions, we analyzed the sequences and structural features of the enzymes. Conserved amino acid residues (FYRDWWN) and putative active sites (N and H) within the DGAT1 subfamily were presented in both large yellow croaker and rainbow trout DGAT1(Supplementary Fig. 2a). One transmembrane domain, potential active site (HPHG) and neutral lipid binding domain residues (FLXLXXXn) were observed in DGAT2 of large yellow croaker and rainbow

trout (Supplementary Fig. 2b). Phylogenetic tree analysis indicated that large yellow croaker and rainbow trout DGAT1 and DGAT2 clustered in teleost clade and evolved separately. DGAT1 and DGAT2 from mammals and rodents were also distinctly separated from the teleost (Fig. 2).

### Tissue expression of *DGAT1* and *DGAT2* in croaker and trout
Further, absolute quantitative PCR was used to determine *DGAT1* and *DGAT2* mRNA expression in different tissues of large yellow croaker and rainbow trout. The highest level of *DGAT1* was detected in the intestine of large yellow croaker and rainbow trout, while the lowest level was observed in the eye of large yellow croaker and the liver of rainbow trout (Fig. 3a, b). *DGAT2* was predominantly expressed in the liver of large yellow croaker, while mainly expressed in the adipose tissue of rainbow trout (Fig. 3c, d). Expression levels of *DGAT1* and *DGAT2* in the liver of large yellow croaker were significantly higher than that in rainbow trout.

### TAG synthesis and lipid droplet biosynthesis of croaker and trout DGATs
To determine the function of DGATs in large yellow croaker and rainbow trout, lcDGATs and omDGATs were expressed in HEK-293T cells and yeast H1246 cells[45,46]. Oleic acid and acetic acid supplementation mimic exogenous fatty acid released from circulating lipoproteins and de novo synthesized fatty acid, respectively. TAG contents were significantly increased when lcDGAT1 and lcDGAT2 were co-expressed in HEK-293T cells in the presence of acetic acid ($P < 0.05$) (Fig. 4a). TAG levels were significantly enhanced in HEK-293T cells transfected with lcDGAT1 or lcDGAT2 in the presence of oleic acid ($P < 0.05$), and this enhancement was more pronounced when co-transfected with lcDGAT1 and lcDGAT2 (Fig. 4b). However, overexpression of either omDGAT1 and omDGAT2 alone, or both together could significantly increase TAG levels in HEK-293T cells incubated with oleic acid or acetate acid ($P < 0.05$) (Fig. 4c, d). In addition, BODIPY 493/503 staining showed that the number of lipid droplets was increased in H1246 cells transfected with lcDGATs or omDGATs (Fig. 4e). These results suggested that DGATs in large yellow croaker and rainbow trout might be involved in TAG synthesis and lipid droplet formation.

### Function and expression of DGATs in croaker and trout under high fat treatment
To examine whether DGAT1 and DGAT2 have the function in hepatic TAG accumulation under high fat condition, primary hepatocytes were incubated with oleic acid (OA) to mimic HFD in vitro. TAG levels were significantly increased in large yellow croaker (171%) and rainbow trout (141%) hepatocytes incubated with OA for 48 h ($P < 0.05$) (Fig. 5a). In large yellow croaker hepatocytes, inhibition of DGAT1 alleviated TAG accumulation induced by OA treatment, but no significant differences in TAG levels were found after treatment with DGAT2 inhibitor ($P < 0.05$) (Fig. 5b). In rainbow trout hepatocytes, inhibition of DGAT1 and DGAT2 together blocked OA-induced increase in TAG levels ($P < 0.05$) (Fig. 5c). The expression of *lcDGAT1* was increased in large yellow croaker hepatocytes after incubation with OA for 6 h, 12 h and 24 h ($P < 0.05$), while expression of *omDGAT1* was not different in rainbow trout (Fig. 5d, e). No significant differences in *DGAT2* mRNA levels were detected in both large yellow croaker and rainbow trout hepatocytes after OA treatment (Fig. 5f, g). In addition, DGAT1 protein levels were increased in large yellow croaker hepatocytes but not in rainbow trout hepatocytes (Fig. 5h, i).

### Identification of transcription factors on *DGAT1* promoters
To identify which transcription factors bind and regulate the *DGAT1* promoter, the binding sites for transcription factors including USF1, USF2, CREBH, CEBPα, CEBPβ, SREBP1, LXR, PPARγ and CHREBP were predicted within the *DGAT1* promoter region of large yellow croaker and rainbow trout. Dual-luciferase reporter assays revealed that USF1, USF2, CREBH and CHREBP significantly enhanced the luciferase activity of croaker *DGAT1* promoter ($P < 0.05$) (Fig. 6a). Conversely, few transcription

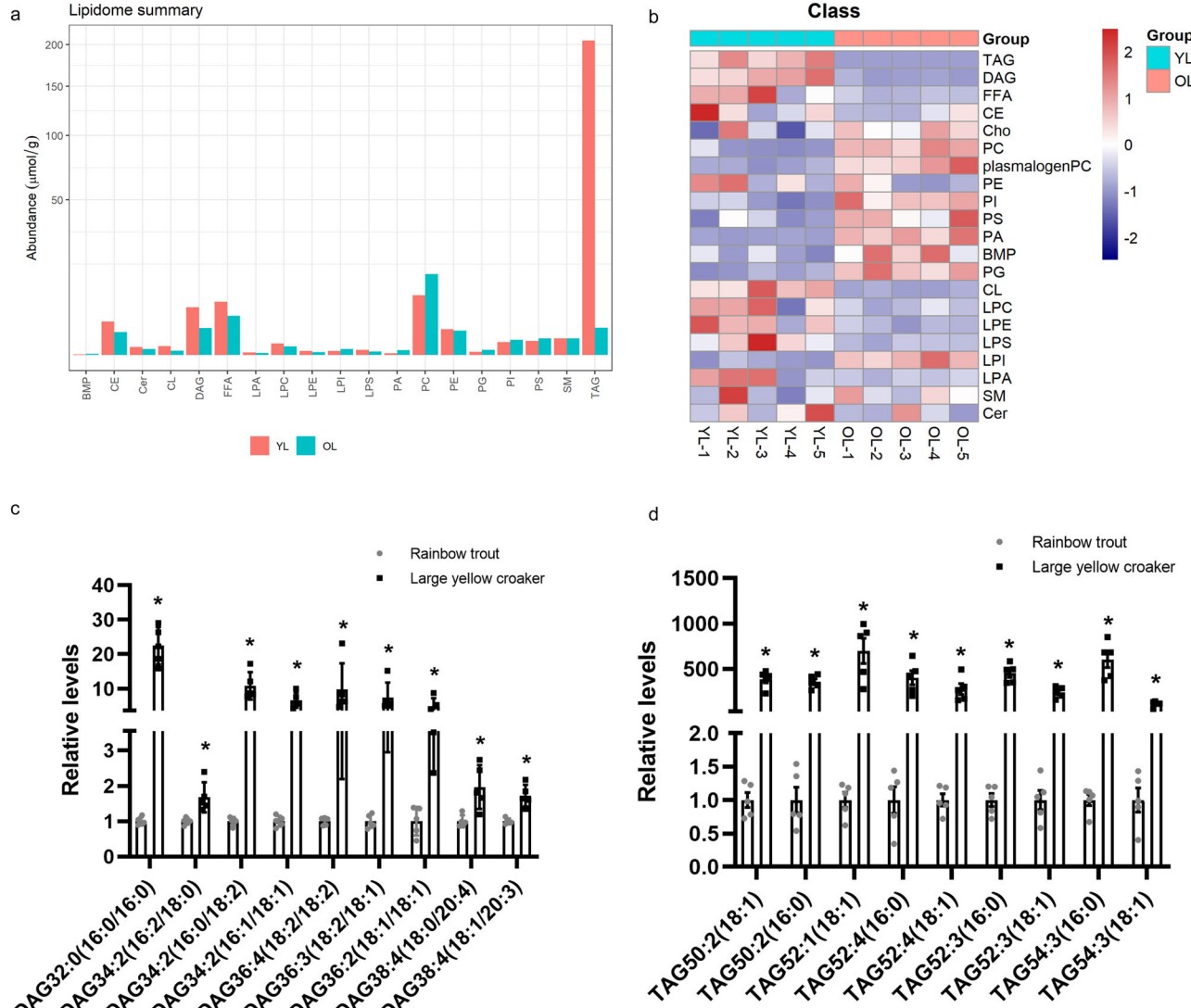

**Fig. 1 | Lipid profiles in livers of large yellow croaker and rainbow trout.**
**a** Contents of each lipid class were shown as μmol/g tissue and represent mean in large yellow croaker (red) and rainbow trout (blue) ($n = 5$). **b** Heat map of lipid classes in livers of large yellow croaker and rainbow trout ($n = 5$). **c-d** Relative contents of specific diacylglycerol species (**c**) and triacylglycerol species (**d**) in livers of large yellow croaker and rainbow trout (*$P < 0.01$, $t$-test; $n = 5$). The content in rainbow trout livers was selected as normalization. Error bars in all figures

represent ± SEM. BMP, bis monoacylglycerol phosphate. CE cholesterolester. Cer Ceramide. CL cardiolipin. DAG diacylglycerol. FFA free fatty acid. LPA lysopho-sphatidic acid. LPC lysophosphatidylcholine. LPE lysophosphatidylethanolamine. LPI lyso phosphatidylinositol. LPS lysophosphatidylserine. PA phosphatidic acid. PC phosphatidylcholine. PE phosphatidyletethanolamine. PG phosphatidylglycerol. PI phosphatidylinositol. PS phosphatidylserine. SM sphingomyelin. TAG triacylglycerol.

factors had significantly effect on the luciferase activity of rainbow trout *DGAT1* promoter (Fig. 6b).

To determine whether these transcription factors are involved in the action of high fat diet, we first quantified their expression in response to excessive OA. It showed that *creb3l3* mRNA was significantly increased in large yellow croaker hepatocytes treated with OA ($P < 0.05$) (Fig. 6c). No significant differences in *chrebp* and *usf2* mRNA expression were observed in large yellow croaker hepatocytes at different points after incubation with OA containing medium (Fig. 6d, e). Expression of *usf1* was downregulated in large yellow croaker hepatocytes treated with OA for 4 h ($P < 0.05$) (Fig. 6f). Consistently, expression of *creb3l3* in the liver of large yellow croaker fed HFD was significantly higher than that in the control group in vivo ($P < 0.05$) (Fig. 6g). In rainbow trout hepatocytes, *creb3l3* mRNA was significantly upregulated after OA incubation for 8 h and 12 h ($P < 0.05$) (Fig. 6h).

**DGAT1 promoter was directly activated by CREBH in croaker**
Next, the regulatory effect of CREBH on large yellow croaker *DGAT1* promoter activity was further performed in HEK-293T cells. The *DGAT1* promoter activity was dramatically increased in HEK-293T cells after transfection with CREBH-N and the degree of promoter activation was significantly higher in comparison with transfection with CREBH ($P < 0.05$) (Fig. 7a). The *DGAT1* promoter activity increased with the increasing of CREBH level (Fig. 7b). Mutation of the predicted CREBH binding site significantly inhibited CREBH induced luciferase activity ($P < 0.05$) (Fig. 7c). Treatment with the potent activator of CREBH (cAMP) could significantly enhance the positive effect of CREBH on the *DGAT1* promoter activity ($P < 0.05$) (Fig. 7d). In addition, ChIP result revealed that the *DGAT1* promoter fragment contained CREBH recognition sites, indicating that CREBH was the sequence-specific binding element to the upstream regulatory region of *DGAT1* (Fig. 7e).

**Fig. 2 | Phylogenetic tree of DGAT1 and DGAT2 in large yellow croaker and rainbow trout.** Sequences were analyzed by the neighbor-joining distance method. The horizontal branch length is proportional to amino acid substitution rate per site. The numbers represent the frequencies (%) with which the tree topology presented was replicated after 1000 iterations.

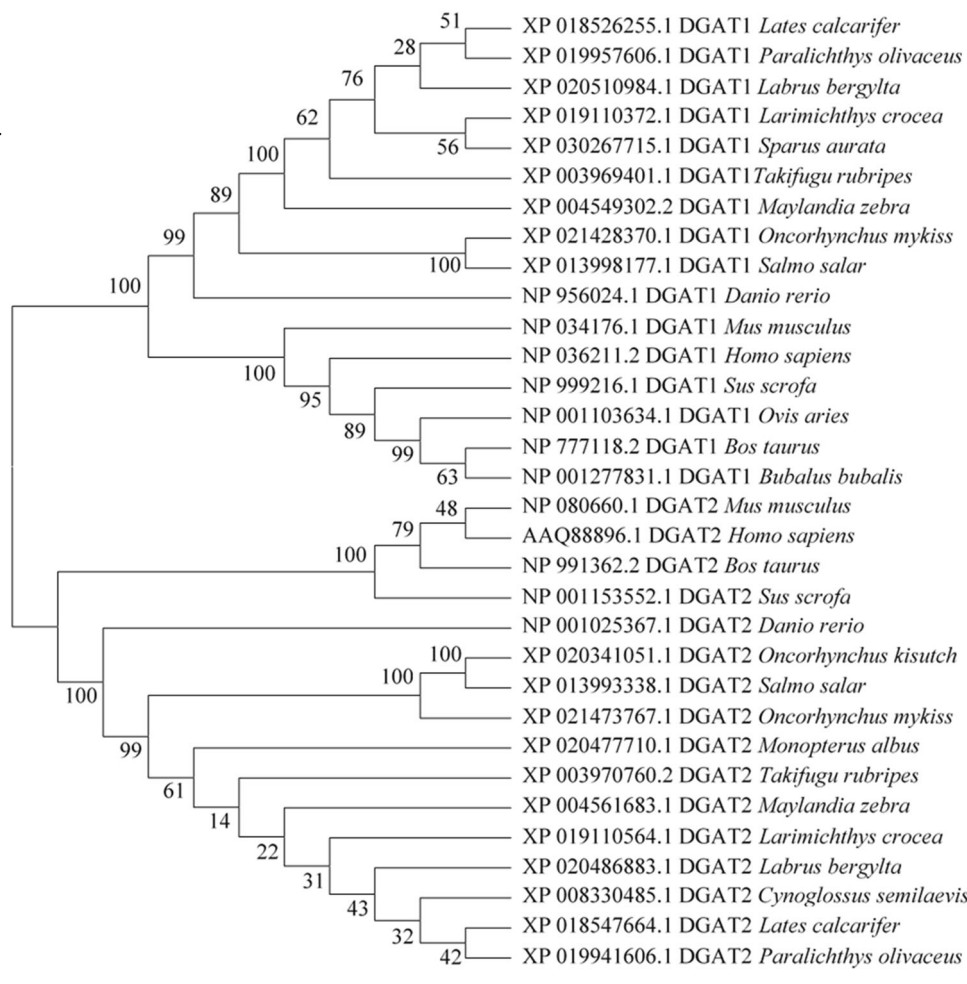

### Regulation of CREBH by GSK3β–mediated phosphorylation

The hepatic CREBH protein expression was increased in large yellow croaker under high fat challenge in vitro (Fig. 8a). However, the increase in CREBH protein level was not observed in rainbow trout (Supplementary Fig. 3a). It was reported that phosphorylation of CREBH by GSK3 accelerates CREBH protein degradation in mammals[47]. We found that phosphorylation of GSK3β at Ser9 was increased in large yellow croaker hepatocytes treated with OA for 4 h (Fig. 8b). Inhibition of GSK3β significantly increased the mRNA level of *creb3l3* and *dgat1* in large yellow croaker hepatocytes (Fig. 8c). The CREBH protein expression was significantly reduced in HEK-293T cells co-transfected with large yellow croaker GSK3β and CREBH (Supplementary Fig. 3b).

To further investigate the impact of GSK3β on the regulation of CREBH, we constructed CREBH mutants (2 S, DSG, 2 S + DSG) in which we substituted conserved serines(S) by alanines(A) (Supplementary Fig. 3c). The protein levels of the corresponding mutant proteins were significantly increased compared to wide type CREBH (WT) (Fig. 8d). The *DGAT1* promoter was significantly activated in HEK-293T cells transfected with 2 S or DSG mutants compare to wide type, and more so by the 2 S + DSG mutant ($P < 0.05$) (Fig. 8e).

### Discussion

In fish, liver plays an important role in lipid storage and the propensity to accumulate hepatic TAG is species-dependent[48]. However, the molecular mechanism underlying hepatic TAG deposition disparity in fish has rarely been investigated. Here, we found that DGATs had conserved functions in TAG synthesis and showed significant inter-species differences in DGATs expression patterns and transcriptional regulation in fish with distinct hepatic TAG deposition, which suggested DGATs could be candidate genes contributing to hepatic TAG deposition disparity in fish.

In this study, we revealed that large yellow croaker and rainbow trout fed the same nutrition diet differed greatly in hepatic TAG level. The results well supported previous studies that considerable differences among fish species exist in hepatic lipid deposition[42]. Previous studies have shown that changes in function and expression of key genes regulating hepatic lipid metabolism contributed to the difference in the hepatic fat content in mammals[4,49,50]. DGATs are critical enzymes for TAG synthesis and have an important role in hepatic TAG synthesis[51–53]. Therefore, we surmised that differences in DGATs between croaker and trout could cause hepatic TAG deposition disparity in two fish species.

Based on the above hypothesis, our data verified that DGAT1 and DGAT2 in croaker and trout were functionally conserved in TAG synthesis. However, substantial differences in absolute mRNA expression patterns of *DGAT1* and *DGAT2* were observed between croaker and trout. Expression of *DGAT1* and *DGAT2* was significantly higher in the liver of croaker compared with trout. These results were consistent with previous reports indicating that inter-individual variations in fat content within the same tissue might be due to differences in the transcript abundance of DGATs[54–56]. Alternatively, previous studies have demonstrated that liver-specific overexpression of DGATs significantly increased liver TAG content in mice and DGATs ASO treatment caused a marked reduction in hepatic TAG level[52,57]. Furthermore, we found that the copy number of DGAT2 was significantly higher in the liver of croaker and in visceral adipose tissue of trout. This was consistent with the primary function for DGAT2 in mediating TAG storage under basal condition[58]. Altogether, these results indicated that DGATs display marked differences in hepatic *DGATs* expression

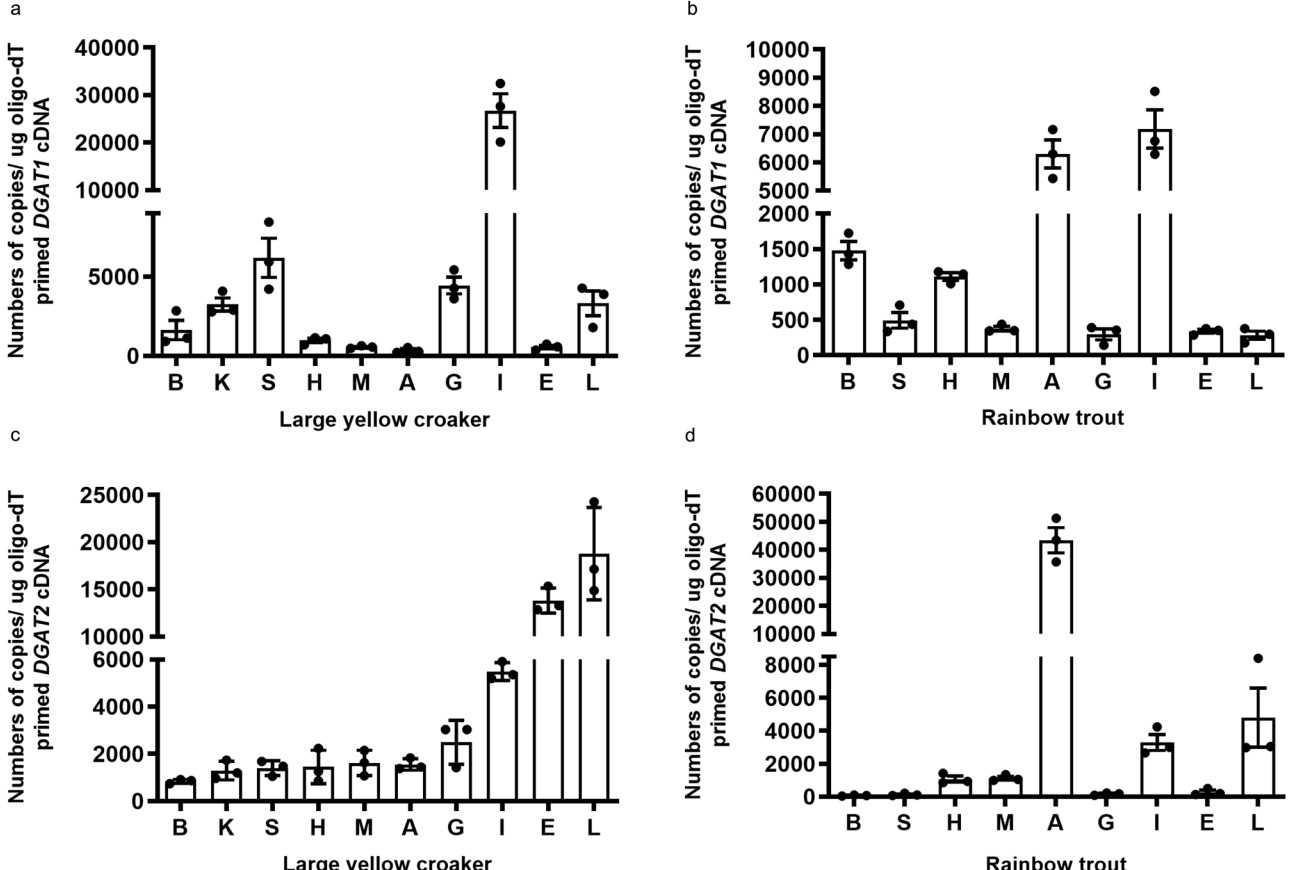

**Fig. 3 | Tissue distribution of *DGAT1* and *DGAT2* in large yellow croaker and rainbow trout.** Expression pattern of *DGAT1* (**a**, **b**) and *DGAT2* (**c**, **d**) in large yellow croaker and rainbow trout. The mRNA expression of genes was analyzed by absolute quantitation. Data were presented as mean ± SEM (*n* = 3). B brain, K kidney, S spleen, H heart, M muscle, A adipose tissue, G gill, I intestine, E eye, L liver.

between croaker and trout, especially DGAT2, which may mediate different hepatic TAG levels in two fishes under physiological condition.

Lipid overload is the major cause of excessive TAG storage in the liver. In this study, we observed that the increase in TAG levels induced by OA in croaker hepatocytes was much higher than that in trout. This result was consistent with previous reports that the degree of excessive hepatic TAG deposition induced by HFD varies extensively among different fish species[6,23,59]. Notably, OA incubation upregulated *DGAT1* expression in croaker hepatocytes rather than in trout. Moreover, we did not observe any significant effect of OA incubation on *DGAT2* expression in two fish species. These results were supported by previous studies showed that DGAT1 played a greater role in esterifying exogenous fatty acids and reducing cellular lipotoxicity[58,60]. Altogether, these results revealed that DGAT1 expression in two fishes showed differential responses to lipid overload that likely contributed to the difference in hepatic TAG accumulation. DGAT2 might play major triglyceride synthesis function in the tissues or species which have strong de novo synthesis capacity.

Differentially expressed genes are more likely to be consequences of distinct regulation mechanisms. In this study, different transcriptional regulations of *DGAT1* promoter by transcription factors were observed. Transcription factors CREBH, USF1, USF2 and ChREBP could up-regulate the promoter activity of *DGAT1* in croaker. Meanwhile, lipid overload increased CREBH expression in vivo and in vitro. These results suggested that CREBH might be the critical transcription factor that regulated the expression of DGAT1 in croaker under OA incubation. Furthermore, CREBH had no effect on the promoter activity of *DGAT1* in trout. Previous studies revealed that CREBH controls the expression of a variety of target genes involved in lipid metabolism and then regulates lipid homeostasis.

To the best of our knowledge, CREBH has positive effect on the *DGAT1* promoter and then regulates hepatic TAG content, which also displays the species specificity. Overall, these results strongly suggested that CREBH has distinct regulatory effects on *DGAT1* promoters between croaker and trout, which may be a contributing factor to differences in hepatic TAG deposition under high fat challenge.

To further clarify the mechanism involved in hepatic TAG deposition, the regulatory mechanism of CREBH protein expression in croaker was investigated. Previous studies suggested that the GSK3 has been identified as one of kinases responsible for the phosphorylation of CREBH and coordinately regulated its protein expression[47]. Our results revealed that OA treatment could increase CREBH protein expression via GSK3β-CREBH axis and then up-regulate *DGAT1* expression in croaker. However, OA incubation had no significant effect on CREBH protein expression in trout. Therefore, these results suggested that differences in the GSK3β-CREBH-DGAT1 axis regulation between croaker and trout might contribute to hepatic TAG deposition disparity in two fish species, which provided a molecular foundation for further investigation into the mechanism of hepatic TAG deposition. Furthermore, these variation in the regulation of DGATs gene might reflect different adaptation strategies of fish to their enormously diverse environments.

In summary, DGATs in two fishes are functionally conserved, but hepatic DGATs expression and transcription regulation varies widely. This study provided a perspective on the use of fish as vertebrate genetic model for studying the mechanism underlying hepatic TAG deposition. Furthermore, the GSK3β-CREBH-DGAT1 axis has the potential to serve as a therapeutic target for the treatment of fatty liver disease and improve the quality and safety of aquatic products.

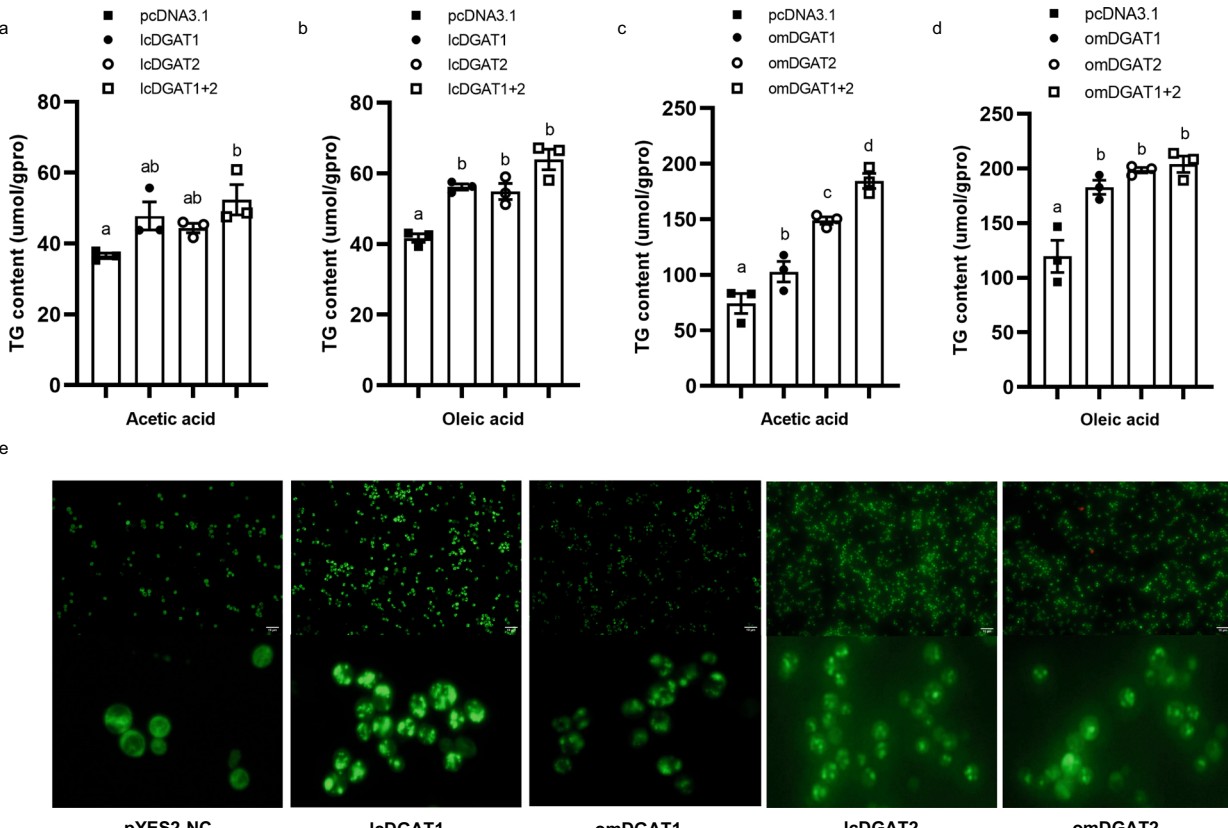

**Fig. 4 | Functional characterization of large yellow croaker (lcDGATs) and rainbow trout (omDGATs) DGATs in HEK-293T cells and yeast H1246 cells.** **a–d** HEK-293T cells overexpressing DGAT1 and/or DGAT2 of large yellow croaker (**a, b**) and rainbow trout (**c, d**) were treated with oleic acid (50 μM) or acetic acid (50 μM) as indicated. After 24 h of incubation, triacylglycerol contents were measured ($n = 3$; mean ± SEM). Means that share the same superscript letter are not significantly different, as determined by Tukey's test ($P > 0.05$). **e** Bodipy staining in H1246 cells by overexpression DGAT1 or DGAT2 of large yellow croaker and rainbow trout. The strain containing the empty pYES2-CT plasmid was used as negative control. Scale bar: 10 um.

## Materials and methods
### Animal experiments and sample collection
All animal experiments were performed in accordance with the Institutional Animal Care and Use Committee of the Ocean University of China (SD2007695). We have complied with all relevant ethical regulations for animal use. Large yellow croaker (10.03 ± 0.02 g) and rainbow trout (10.07 ± 0.03 g) juveniles were fed with the same nutritional diet (43% crude protein and 12% crude fat) and cultured for 10 weeks. The feeding trial protocol was described by Li et al. and the fishes showed no significant difference in physical performance[61,62]. After 10 weeks feeding trial, liver tissues were snap frozen by liquid nitrogen immediately and were stored at −80 °C before lipidomic profiling.

Two isoproteic (43% crude protein) diets were prepared to contain different levels of dietary lipid (12% and 18% on a dry basis) and named as control group (CON) and high fat diet (HFD). Each diet was randomly assigned to cages in triplicate. The fish were reared in floating sea cages (3.0 × 3.0 × 3.0 m) for 2 weeks for acclimation to the experimental conditions. Before the feeding trial, fish were fasted for 24 h and weighed. Large yellow croaker juveniles of similar size (12.00 g ± 0.10 g) obtained from Ningbo, China, were reared in floating sea cages at 24–28°C. The feeding trial protocol was described by Wang et al. [10] The fish were hand-fed to apparent satiation twice daily (05:00 and 17:00). After 10 weeks feeding trial, liver tissues were collected from six individuals in each cage and frozen immediately in liquid nitrogen.

### Lipidomic analyses
Lipids were extracted from large yellow croaker liver (YL) and rainbow trout liver (OL) (60 mg) using a modified Bligh and Dyer extraction procedure (double rounds of extraction). Lipid extracts were collected into a single tube and dried in the SpeedVac under OH mode. Samples were stored at −80 °C until processed. All lipidomic analyses were performed on an Exion LC-system coupled with a QTRAP 6500 PLUS system (Sciex, Framingham, USA), and individual lipids from various classes were quantitated as described previously[63].

### RNA extraction, cDNA synthesis and molecular cloning
Total RNA extraction was carried out using the RNAiso Plus Reagent (TaKaRa, Japan) according to the manufacturer's protocols. All RNA samples were verified on agarose gel electrophoresis and the ratios of A260/A280. Total RNA (1 μg) was treated with RNase-free DNase I and was reverse-transcribed using PrimeScriptTM RT reagent kit (Takara, Japan). The first pair of amplification primer designed according to the predicted sequence in large yellow croaker and rainbow trout genomes. All the primers were listed in Supplementary Table 1.

### Sequence, phylogenetic analysis and bioinformatic analysis
Sequences alignments were analyzed based on the BLAST program of National Center for Biotechnology Information (http://www.ncbi.nlm.nih.gov). Amino acid sequence alignments were carried out using DNAMAN. One neighbor-joining phylogenetic tree was constructed using MEGA 6.0. Protein domains were identified and analyzed using SMART.

### Absolute and relative mRNA quantification
Quantitative real-time PCR (qRT-PCR) primers were designed by Primer Premier 5.0 based on cloned *DGATs* gene sequences. cDNA was diluted 7-fold using RNase- and DNase-free water. qRT-PCR was performed in a

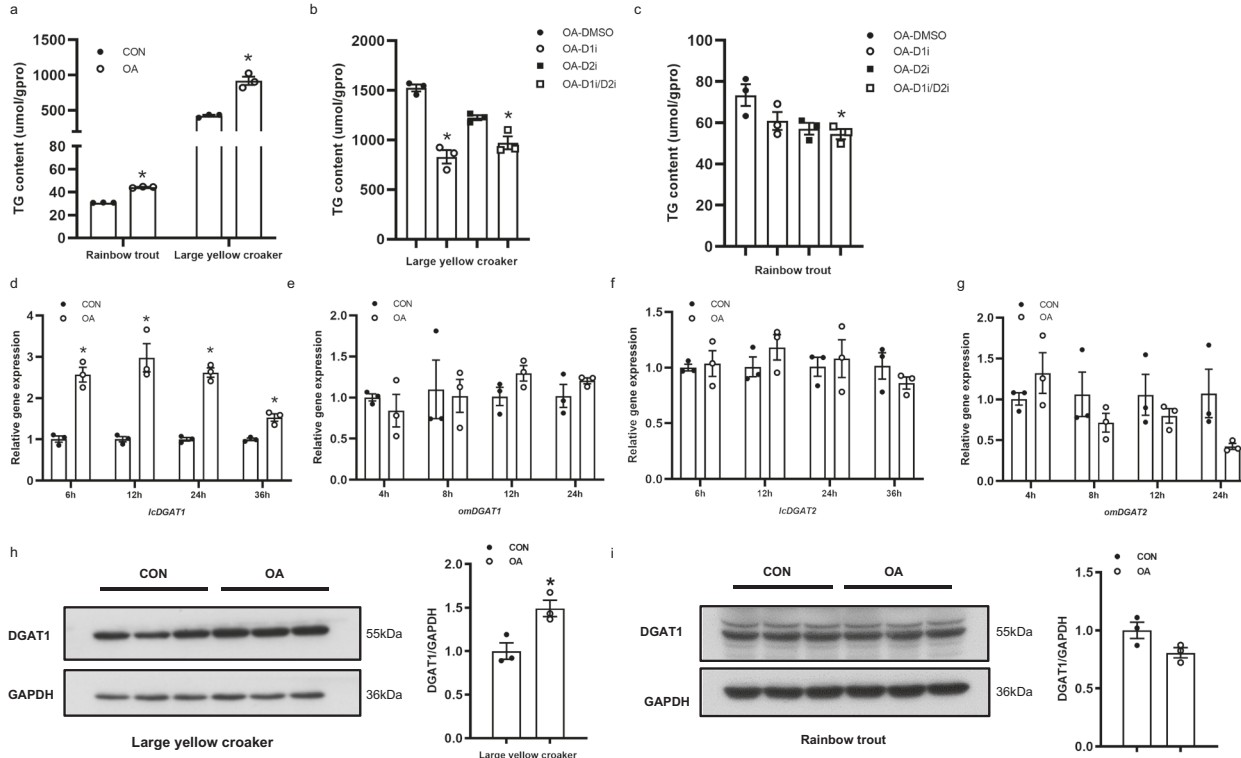

**Fig. 5 | Function and expression of *DGATs* under high-fat challenge. a** TAGs levels in large yellow croaker and rainbow trout hepatocytes incubated with 800 uM oleic acid (OA) (*n* = 3). **b**-**c** Effects of DGATs inhibitors on triglyceride contents in large yellow croaker (**b**) and rainbow trout (**c**) hepatocytes treated with OA (*n* = 3). **d**–**g** Relative mRNA expression of *DGAT1* (**d**, **e**) and *DGAT2* (**f**, **g**) in large yellow croaker and rainbow trout hepatocytes at different time points after OA treatment (*n* = 3). **h**–**I** Western blot analysis of DGAT1 during OA incubation for 24 h in hepatocytes of large yellow croaker (**h**) and rainbow trout (**i**) (*n* = 3). Error bars represent ± SEM, *\*P* < 0.05.

quantitative thermal cycler (Mastercycle ep realplex, Eppendorf, Hamburg, Germany) in a total volume of 20 µL. At the end of PCR amplification, melting curve analysis was performed to confirm that only one PCR product was present.

Absolute quantification qPCR is a novel quantification method without the requirement of reference gene[64]. In general, primers were designed to span intron–exon boundaries and included the fragment of relative expression mRNA. The PCR products were cloned into a T7 vector and subsequently purified as Plasmid DNA. Linearization plasmids were quantified via the TB Green quantification method at 1:10 serial dilutions. A standard curve was drawn by plotting the natural log of the threshold cycle (CT) against the natural log of the number of molecules. The quantification results were presented as copy number per microgram of oligo-dT primed cDNA.

### Functional characterization of DGATs in HEK-293T cells and yeast H1246 cells

DGATs coding sequence (CDS) of large yellow croaker and rainbow trout were amplified with primers containing restriction sites *BamH* and cloned into pcDNA3.1+ expression vector. The pcDNA3.1-DGATs plasmids (2ug) were transfected into HEK-293T cells at 80% confluence (6-well, $2 \times 10^6$ cells/mL) using lipofectamine 3000 reagent (Invitrogen, USA) according to the manufacturer's instructions. Cellular lysates were collected 48 h after transfection using cell lysis buffer in the TAG Assay Kit (Pulilai, Beijing, China). DGAT1 and DGAT2 inhibitor (T863 and PF-06424439) are purchased from Sigma. Triglycerides were measured by a glycerol lipase oxidase (GPO-PAP) method according to the TAG reagent kit instructions.

DGATs CDS of large yellow croaker (lcDGAT) and rainbow trout (omDGAT) were cloned into pYES2-CT vector (Invitrogen, USA) to construct plasmid pYES2-lcDGAT1, pYES2-lcDGAT2, pYES2-omDGAT1, and pYES2-omDGAT2, separately. *Saccharomyces cerevisiae* H1246 (Δ*dga1* Δ*lro1* Δ*are1* Δ*are2*), a triglyceride synthesis-deficit mutant kindly donated by Professor Sten Stymne at Swedish University of Agricultural Sciences and Professor Jin Liu at Peking University. pYES2-CT and pYES2-DGATs plasmids were individually transformed into yeast H1246 cells using the LiAc method. The transformants were selected on synthetic complete medium plates lacking uracil and containing 2% (w/v) glucose. A single colony was chosen and grown in selective medium overnight. The yeast cells were harvested by centrifugation at 1500 x g for 5 min and were then diluted to an OD600 of 0.4. When the OD600 of the cultures reached 0.8–1.0, cultures were shifted to induction medium (SC-uracil medium containing 2% galactose). After 48 h, cells were collected by centrifugation, washed twice with pre-cooling sterile water and incubated with BODIPY 493/503 (Invitrogen, USA) for 15 min in the dark. Samples were washed with phosphate buffer saline three times. Images were obtained using fluorescence microscope (Nikon, ECLIPSE 80i).

### Primary hepatocytes culture

Liver tissues were separately obtained from large yellow croaker and rainbow trout after starved for 24 h. Primary hepatocytes were isolated using trypsin methods[61,62]. In general, livers from croaker are collected randomly and are cut into small pieces ( ~ 1mm³). After cleaning, liver tissue pieces are digested by 0.25% trypsin for 10–15 min, and then were gently filtered through a 100 um mesh. The cells were harvested by centrifugation and treated with red cell lysis buffer. After washing, hepatocytes were plated in 6-well plates ($2 \times 10^6$ cells/mL) in DMEM/F12 media containing 15% fetal bovine serum (BI, Israel) at 27 °C and 17 °C, respectively. Adherent cells were treated with serum-free medium for 3 h and then incubated with 800 µM OA conjugated with 2% fatty acid-free BSA for different time points. Cells were collected for further gene expression and lipid analysis.

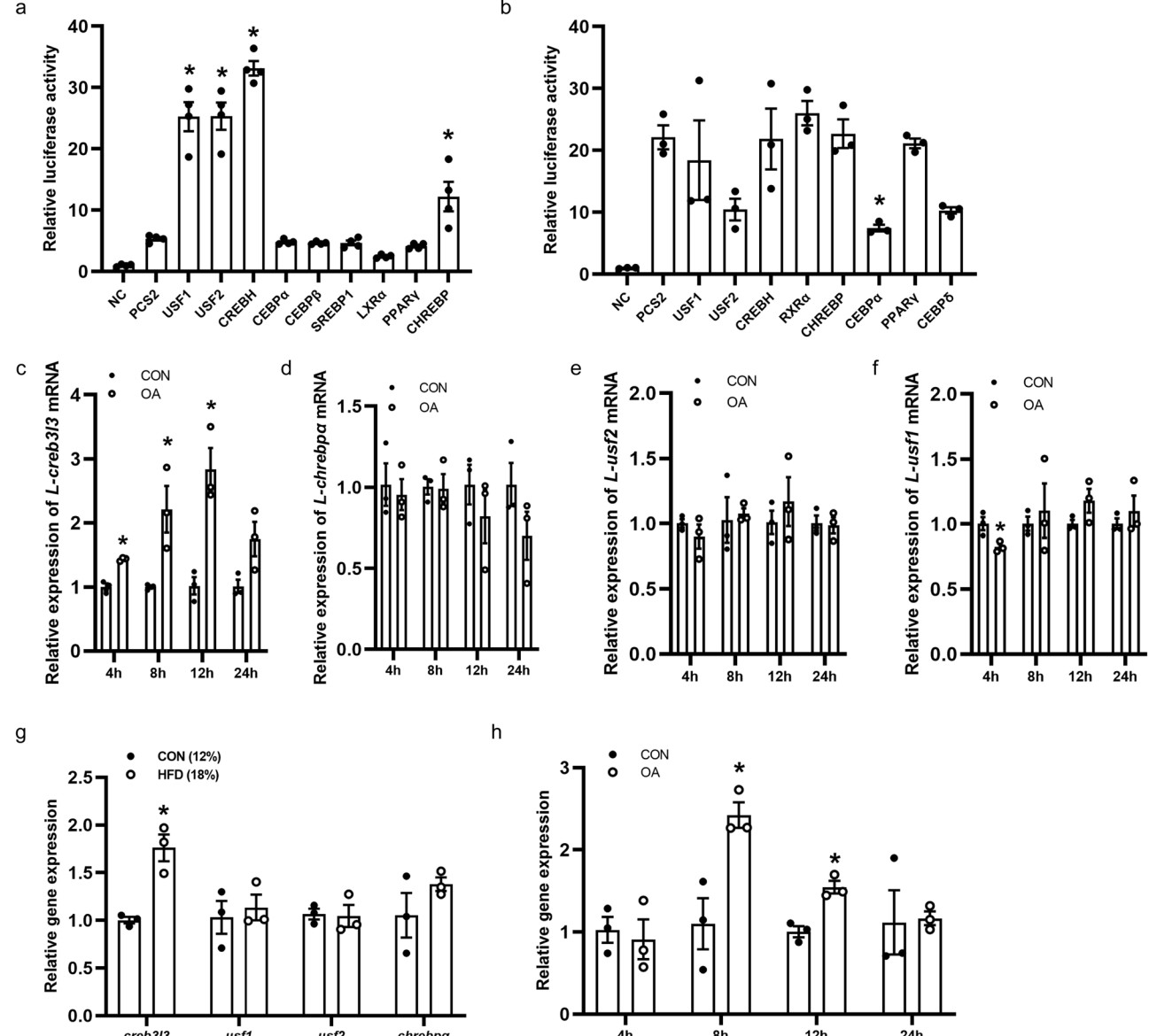

**Fig. 6 | CREBH activated the *DGAT1* promoter in large yellow croaker in response to high fat challenge. a, b** Effects of transcription factors on the *DGAT1* promoter of large yellow croaker (**a**) and rainbow trout (**b**) in HEK-293T cells. The negative control (NC) was an empty plasmid with no promoter sequence (pGL6-basic) and empty PCS2 plasmid which was used to normalize (*n* = 3/4). The *DGAT1* promoter was co-transfected with plasmids of transcription factor in HEK-293T cells. **c–f** Relative mRNA expression of *l-creb3l3* (**c**), *l-chrebp* (**d**), *l-usf2* (**e**) and *l-usf1* (**f**) in hepatocytes of large yellow croaker (*n* = 3). Cells treated with BSA was the control group. **g** mRNA levels of *l-creb3l3*, *l-usf2*, *l-chrebp* and *l-usf1* in livers of large yellow croaker (*n* = 3). **h** mRNA levels of *o-creb3l3* in rainbow trout hepatocytes treated with oleic acid. Data are expressed as mean ± SEM. *$P < 0.05$.

RNA extraction, cDNA synthesis and qRT-PCR were carried out as previous described. All primer sequences were list in Supplementary Table 1. β-actin was suggested as control gene for qRT-PCR through the significance analysis of NormFinder algorithms and geNorm. Data were calculated using the $2^{-\Delta\Delta Ct}$ method.

**Cloning of *DGAT1* promoter and Dual-Luciferase Reporter Assays**

Genomic DNA from large yellow croaker and rainbow trout were extracted as template to amplify *DGAT1* promoters. Promoters of *DGAT1* were cloned into pGL6-basic plasmid with a luciferase reporter. Putative transcription factor binding sites on *DGATs* promoter were predicted by using JASPAR and TF Binding (http://tfbind.hgc.jp/). Expression plasmids of USF1, USF2, SREBP1, PPARγ, LXRα, RXRα, ChREBP, CEBPα and CEBPβ were previously obtained in our laboratory[42,43]. CDSs of *creb3l3* in large

yellow croaker (GenBank: XM_010733027.2) and rainbow trout (GenBank: XM_021622081.1) were inserted into PCS2+ vector using the *XhoI* restriction site. The N-terminal fragment of CREBH (CREBH-N) (amino acids 1-291) was amplified from the full length of CREBH (CREBH-F). PCR primers for various plasmid construction was listed in Supplementary Table 2. Mutation of the CREBH binding site on the *DGAT1* promoter luciferase construct (*DGAT1*-mut) was carried out by site-directed mutagenesis.

Dual-Luciferase Reporter Assays were performed through co-transfecting the DGATs reporter plasmid (0.2 μg/μL), transcription factor plasmids (0.2 μg/μL) and pRL-TK renilla luciferase plasmid (0.04 μg/μL) into HEK-293T cells. Assays were performed 24 h after transfection using the Dual-Luciferase Reporter Assay System (TransGen Biotech Co., Ltd, Beijing, China). Data were collected on InfiniTE200 plate reader (Tecan, Switzerland).

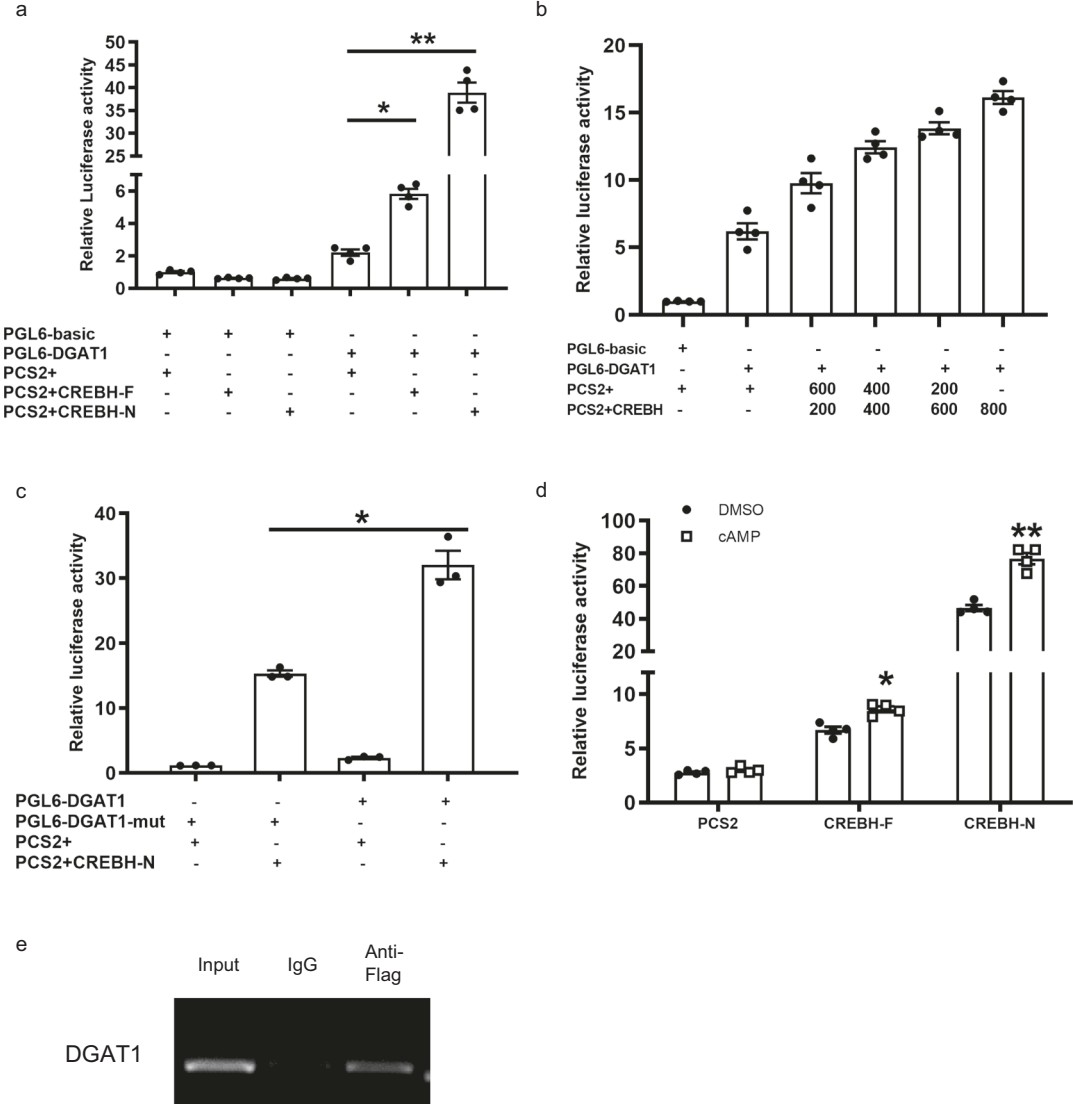

**Fig. 7 | CREBH directly regulated the *DGAT1* promoter in large yellow croaker.**
**a** Relative luciferase activities of the *DGAT1* promoter in large yellow croaker in HEK-293T cells. The control group was co-transfected with pGL6-basic plasmid and pCS2+ empty plasmid. The luciferase activity in control group was selected as normalization ($n = 4$). CREBH-F: CREBH full length. CREBH-N: CREBH nucleus form. **b** Dual luciferase activity analysis of CREBH gradient concentration in the *DGAT1* promoter in large yellow croaker ($n = 4$). **c** Effects of CREBH on the *DGAT1* promoter with mutation in the CREBH binding site ($n = 3$). **d** cAMP stimulation enhanced activation effect of CREBH or CREBH-N on the *DGAT1* promoter ($n = 4$). **e** ChIP analysis of CREBH binding to the *DGAT1* promoter (1709bp~1720bp) in HEK-293T cells. Values represent mean ± SEM. *$P < 0.05$; **$P < 0.01$.

## Chromatin immunoprecipitation (ChIP) assay

HEK-293T cells were co-transfected with PCS2+Flag-CREBH plasmid and PGL6-*DGAT1* promoter vector. After 48 h of transfection, cells were fixed with 1% formaldehyde at 37 °C for 10 min. ChIP assays were performed using the ChIP Kit (Thermo Fisher Scientific) according to manufacturers instructions. Immunoprecipitation was performed using anti-Flag and IgG antibody.

## Immunoblotting analysis

Western blot analysis was conducted as previously described by Zhu et al.[65] Briefly, protein concentrations were determined by a Bradford Protein Assay Kit (Beyotime Institute of Technology, China). Equal amounts of protein (10 μg per lane) were separated by a 10% SDS-PAGE gel and transferred onto activated polyvinylidene fluoride (PVDF) membranes. Membranes were blocked with 5% non-fat milk in TBST (Tris-buffered saline with 0.05% Tween 20) for 2 h at room temperature and incubated with primary antibodies, followed by incubation with horseradish peroxide (HRP)-conjugated secondary antibody. Blots were then visualized using an

enhance chemiluminescence detection reagent (Beyotime Institute of Technology, China). All the antibodies have been tested and showed the right size for target proteins. The membranes were cut while we used cassette and X-Ray films for blots imaging. After scanning, cropped blots for each target protein were saved so that completely uncropped blots are not available. Primary antibodies contain anti-Ser9-GSK3β (1:1000, Cell Signaling Technology, USA), anti-GSK3β (1:1000, Cell Signaling Technology, USA), anti-DGAT1 (1:5000, GenScript, China), anti-CREBH (1:2000, Abcam) and GAPDH (1:5000, Golden Bridge Biotechnology, China).

## Statistics and reproducibility

All the experiments were performed in triplicate and three independent repeats. Data were presented as mean ± SEM using GraphPad Prism 6.0 unless stated otherwise. Abundance of lipid species and classes was compared using Mann-Whitney U test. Statistical significance was analyzed by one-way ANOVA (more than two groups) and independent *t*-test using SPSS 22.0. Data were considered statistically significant if $P < 0.05$.

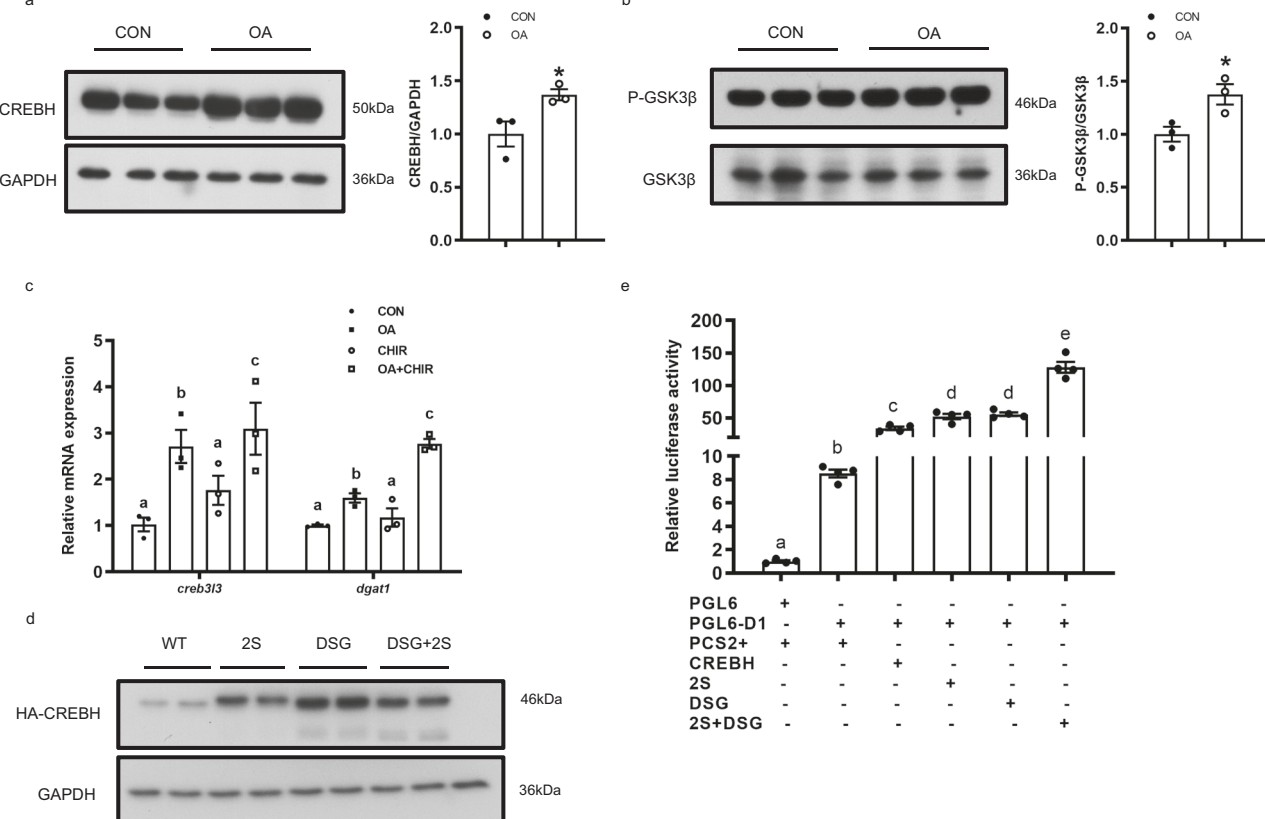

**Fig. 8 | Regulation of CREBH by GSK3β–mediated phosphorylation.**
**a** Immunoblot of CREBH in croaker hepatocytes treated with OA ($n = 3$). **b** Levels of total and phosphorylated GSK3β in croaker hepatocytes were examined by western blot analysis ($n = 3$). **c** mRNA levels of *creb3l3* and *DGAT1* in large yellow croaker hepatocytes treated with vehicle (DMSO) and GSK3β inhibitor (CHIR98014)

($n = 3$). **d** Western blot analysis of protein lysates from HEK-293T cells transfected with CREBH wild type and mutants. **e** The effects of CREBH wide type and mutations on activation of the *DGAT1* promoter ($n = 4$). Means that share the same superscript letter are not significantly different, as determined by Tukey's test ($P > 0.05$). *$P < 0.05$.

## Data availability

The data supporting the findings of this study are available within the Article and its Supplementary Information files. The source data for the graphs are available as an Excel file in Supplementary Data 1. Supplementary Figs. 4–5 contain cropped and unedited blot images. All other data are available from the corresponding author upon reasonable request.

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

## Acknowledgements

We thank Professor Sten Stymne at Swedish University of Agricultural Sciences and Professor Jin Liu at Peking University for sharing the yeast H1246 cells used in this study. Funding: This research was supported by the National Science Fund for Distinguished Young Scholars of China (Grant no. 31525024), Key Program of National Natural Science Foundation of China (Grant no.31830103), Ten-thousand Talents Program (grant no: 2018-29) and the Agriculture Research System of China (Grant no: CARS-47-11).

## Author contributions

X.J.X., R.J. and Q.A. designed research. X.J.X., R.J., S.H., and X.X. performed research performed experiments and analyzed the data. X.J.X., Y.L., S.Z. and J.D. interpreted the data and participated in conceptualizing. X.J.X., R.J., and Q.A. wrote the paper. K.M. and Q.A. supervised the project and provided funding for the studies.

## Competing interests

The authors declare no competing interests.
