## [Peer Review File · Communications Biology]

Reviewers' comments:

Reviewer #1 (Remarks to the Author):

The manuscript proposes that differences in diacylglycerol acyltransferases expression patterns and regulation cause distinct hepatic triglyceride deposition in fish. The mechanism is that DGATs have higher expression on their major position for lipid storage and the promoter of croaker DGAT1 can be upregulated by CREBH, but not in trout.

In general, the paper is well done. Their study provided meaningful and innovative regulatory mechanism for nutrition and metabolism in fish, and probably also in other vertebrates. However, there are several issues in the paper which should be addressed before the acceptance.

General comments:

1. Does other fishes have similar expression pattern of DGATs which might be related with the position for lipid storage?
2. The nutrition requirement for croaker and trout is different. How was the physical performance if they were fed with same diet?
3. In the manuscript, it seems like that DGAT1 can use exogenous fatty acid to synthesize TG and DGAT2 didn't show great function in fish. But DGAT2 is the major enzyme for TG synthesis. How to explain it?

Specific questions:

1. CREBH also can be regulated by the glucose level. It might imply that oleic acid has effect on glucose metabolism and upregulated CREBH expression. It will be better to show some data on glucose.
2. What happens to CREBH expression pattern in croaker and trout?
3. It will be better to show the website for the promoter prediction.
4. Please show the position of DGAT1 promoter for CHIP.

Reviewer #2 (Remarks to the Author):

This manuscript describes the regulation of DGAT expression in large yellow croaker and rainbow trout, which exhibit different rates of hepatic TAG deposition. Findings indicate that differential regulation of DGAT genes contributes to species-specific rates of lipid deposition. Additional experiments identify specific mechanisms of DGAT regulation in large yellow croaker. These findings provide valuable information about nutrient-gene interactions in fish that can contribute to the development of feeding strategies that improve nutrient partitioning and growth efficiency. However, there are some areas of concern that the authors should consider during the revision process.

Title:

1. My recommendation is to improve title specificity by indicating the study was performed in fish

Results/discussion:

2. The bioinformatics section in lines 116-136 does not contribute significantly to the overall findings. Genes have also been annotated and sequences are available in public databases. It is recommended to

remove the bioinformatic analyses described in this section and instead focus on the novel findings that advance knowledge of gene regulation.

3. It is interesting how the expression of hepatic DGAT2 is greater than DGAT1 in both species (Fig 3), yet the protein appears to contribute minimally to oleic acid- and acetic acid-induced TAG formation or lipid droplet formation when overexpressed in cells (Fig 4). Supporting this observation is the OA induced expression DGAT1 but not DGAT2, and inhibition of DGAT2 does not significantly attenuate the OA response (Fig 5). The authors do not address these counter-intuitive findings in the discussion, but it is worthwhile to do so. Can they speculate on the functional significance of such a highly expressed gene (DGAT2) when the functional assays support the lower expressed DGAT1 is more functionally active as a regulator of TAG synthesis? The sentence in lines 261-264 that states DGAT2 is perhaps more significant for species-specific differences should also be reconsidered, given that DGAT1 appears to be more functionally significant than DGAT2.

4. Absent throughout the results is a densitometric analysis of protein abundance for the Western blot assays. It is critical for protein abundance to be quantified and statistically analyzed for differential expression. Currently, evaluation of relative protein abundance can be only a subjective evaluation of how dark/thick the bands are. Given that a considerable number of inferences are made from these data, the findings generated from Western blot analyses must be supported by the appropriate analyses. Similarly, a quantitative analysis of lipid droplet number and/size is essential for images in Fig 4e.

5. Why are t-tests used in some graphs while multiple means comparisons are used other times? For example, Fig 4a uses a multiple means comparison while Figs 4b-d, despite showing the same type of analysis/data, use a t-test to compare to the control. Please be consistent. In cases where comparing between all treatment groups is valuable (such as Fig 4 or Fig 6a,b), then a multiple means comparison will be most insightful regarding relative contribution of each protein to total activity.

6. There should be some statistical support provided to support the statement in line 203 about the dose dependent increase in promoter activity. A simple linear regression will suffice.

Minor comments:

1. Acetate acid should be written as acetic acid.

2. No where in the manuscript is it indicated what inhibitors of DGAT1 and DGAT2 were used.

3. Please improve clarification in the figure legends regarding what two treatments are being compared for t-tests (ie: what treatment is used for the control). In some figures like Fig 6a,b, it is not always obvious.

4. Fig 7: Does CREBH-F mean the full-length transcript, compared to CREBH-N which is the N-terminus of the protein? Some panels in Fig 7 have CREBH while others have CREBH-F. Are they the same thing? Please clarify.

Dear Reviewers,

We thank you for your feedback and comments on our manuscript. Please find our responses to your comments below and revisions to our manuscript in red font.

Response to Reviewer #1's comments:

1. The manuscript proposes that differences in diacylglycerol acyltransferases expression patterns and regulation cause distinct hepatic triglyceride deposition in fish. The mechanism is that DGATs have higher expression on their major position for lipid storage and the promoter of croaker DGAT1 can be upregulated by CREBH, but not in trout. In general, the paper is well done. Their study provided meaningful and innovative regulatory mechanism for nutrition and metabolism in fish, and probably also in other vertebrates. However, there are several issues in the paper which should be addressed before the acceptance.

Response: We appreciate the kind words of Reviewer 1 about our intriguing observations described in the manuscript. In our revised manuscript, we have revised our several issues to strengthen the finding. This is the summary of the data we have added to our revised manuscript:

2. Does other fishes have similar expression pattern of DGATs which might be related with the position for lipid storage?

Response: Thank you for your kind suggestion. We know the position for lipid storage across tissues is highly diverse in fish according to Ren and Xu (1,2)'s results. The correlation analysis is unclear for each species. In the manuscript, we choose croaker and trout as the subject to figure out the mechanism of hepatic triglyceride deposition. It showed that DGATs have different expression pattern. It will be very helpful to check whether other species have similar pattern for the next step. More corresponding work would be performed in the future.

1) Ren W, Li J, Tan P, et al. Lipid deposition patterns among different sizes of three commercial fish species[J]. Aquaculture Research, 2018, 49(2): 1046-1052.

2) Xu H, Bi Q, Pribytkova E, et al. Different lipid scenarios in three lean marine teleosts having different lipid storage patterns[J]. Aquaculture, 2021, 536: 736448.

3. The nutrition requirement for croaker and trout is different. How was the physical performance if they were fed with same diet?

Response: Thanks for the question. We agree with Reviewer 1 that croaker and trout might show potential difference in physical performance. However, it doesn't show any significant difference based on our previously published physical studies (3,4). The growth performance and survival rates have been tested and showed no significant difference. In addition, the diet we designed is suitable for croaker and trout's nutrition requirement. We have included the statement in our revised manuscript.

3) Dong X, Tan P, Cai Z, et al. Regulation of FADS2 transcription by SREBP-1 and PPAR- α influences LC-PUFA biosynthesis in fish[J]. Scientific reports, 2017, 7(1): 40024.

4) Li Y, Tocher D R, Pang Y, et al. Environmental adaptation in fish induced changes in the

regulatory region of fatty acid elongase gene, *elovl5*, involved in long-chain polyunsaturated fatty acid biosynthesis[J]. *International Journal of Biological Macromolecules*, 2022, 204: 144-153.

4. In the manuscript, it seems like that DGAT1 can use exogenous fatty acid to synthesize TG and DGAT2 didn't show great function in fish. But DGAT2 is the major enzyme for TG synthesis. How to explain it?

Response: Thanks for the suggestion. As we know, DGAT1 and DGAT2, are the main enzymes responsible for TAG synthesis. These enzymes do not share high DNA or protein sequence similarities, and it has been suggested that they play specific roles in different tissues and in some species in TAG synthesis, especially on substrate specificity. There is plenty of evidence in mammal researches that DGAT1 and DGAT2 mediate distinct hepatic functions. DGAT2 is primarily responsible for incorporating endogenously synthesized Fatty acids (FAs) into TG, whereas DGAT1 plays a greater role in esterifying exogenous FAs to glycerol.

Teleost fish utilize glucose poorly and the capacity of the pathway to convert glucose into cellular lipids for storage is relatively low (5,6). Fish meal and fish oil are the most popular energy resource for the fish. It makes sense that DGAT1 and DGAT2 are active and DGAT1 has stronger effect in fish liver which utilize lots of exogenous FAs. It is consistent that DGAT1 might be very important for hepatic TG synthesis in fish.

5) Viegas I, Jarak I, Rito J, et al. Effects of dietary carbohydrate on hepatic de novo lipogenesis in European seabass (*Dicentrarchus labrax* L.) [J]. *Journal of lipid research*, 2016, 57(7): 1264-1272.

6) Bou M, Todorčević M, Torgersen J, et al. De novo lipogenesis in Atlantic salmon adipocytes[J]. *Biochimica et Biophysica Acta (BBA)-General Subjects*, 2016, 1860(1): 86-96.

5. CREBH also can be regulated by the glucose level. It might imply that oleic acid has effect on glucose metabolism and upregulated CREBH expression. It will be better to show some data on glucose.

Response: Thanks for the suggestion. According to the advice, we also measured glucose level after OA treatment. It showed cellular glucose level increased. However, it is hard to make a conclusion that it is primary effect or secondary. Because CREBH also can regulate control hepatic glucose and homeostasis. Then we didn't include the data in the manuscript.

6. What happens to CREBH expression pattern in croaker and trout?

Response: Thanks for the suggestion. Based on our result that CREBH can upregulate DGAT1 in

large yellow croaker. We tested the expression pattern of creb313 in two size large yellow croaker. It showed that creb313 has high expression level in croaker liver and intestine.

7. It will be better to show the website for the promoter prediction.

Response: Thanks for the suggestion. Online software JASPAR (<http://jaspar.genereg.net/>) and TF Binding (<http://tfbind.hgc.jp/>) were used to predict potential transcription factor binding sites in the promoter regions of DGATs. We already added the information in the revised manuscript.

8. Please show the position of DGAT1 promoter for CHIP.

Response: Thanks for the suggestion. The binding position for CREBH in DGAT1 promoter is around 1709bp~1720bp. We included the information in the revised manuscript.

Response to Reviewer #2's comments:

1. This manuscript describes the regulation of DGAT expression in large yellow croaker and rainbow trout, which exhibit different rates of hepatic TAG deposition. Findings indicate that differential regulation of DGAT genes contributes to species-specific rates of lipid deposition. Additional experiments identify specific mechanisms of DGAT regulation in large yellow croaker. These findings provide valuable information about nutrient-gene interactions in fish that can contribute to the development of feeding strategies that improve nutrient partitioning and growth efficiency. However, there are some areas of concern that the authors should consider during the revision process.

Response: We thank this reviewer for his/her critical and helpful evaluation of our manuscript. We hope the results can help design feeding strategies that improve nutrient partitioning and growth efficiency. In response to the reviewer's critique, our manuscript has undergone a major revision.

2. My recommendation is to improve title specificity by indicating the study was performed in fish

Response: Thanks for the advice. We agreed with the suggestion and revised our title as “Differences in diacylglycerol acyltransferases expression patterns and regulation cause distinct hepatic triglyceride deposition in fish”.

3. The bioinformatics section in lines 116-136 does not contribute significantly to the overall findings. Genes have also been annotated and sequences are available in public databases. It is recommended to remove the bioinformatic analyses described in this section and instead focus on the novel findings that advance knowledge of gene regulation.

Response: We thank the reviewer for this comment and agree. According to the reviewer’s suggestion, we simplified the bioinformatic section and deleted some unnecessary sequence analysis. And phylogenetic and motif analysis of DGATs were included in the revised manuscript because it showed that that amino acid sequence of DGATs was generally conserved in the evolution and their function should be conserved. It is consistent with our hypothesis that the function of DGATs in fish are conserved and the gene regulation are different.

4. It is interesting how the expression of hepatic DGAT2 is greater than DGAT1 in both species (Fig 3), yet the protein appears to contribute minimally to oleic acid- and acetic acid-induced TAG formation or lipid droplet formation when overexpressed in cells (Fig 4). Supporting this observation is the OA induced expression DGAT1 but not DGAT2, and inhibition of DGAT2 does not significantly attenuate the OA response (Fig 5). The authors do not address these counter-intuitive findings in the discussion, but it is worthwhile to do so. Can they speculate on the functional significance of such a highly expressed gene (DGAT2) when the functional assays support the lower expressed DGAT1 is more functionally active as a regulator of TAG synthesis? The sentence in lines 261-264 that states DGAT2 is perhaps more significant for species-specific differences should also be reconsidered, given that DGAT1 appears to be more functionally significant than DGAT2.

Response: Thanks for the suggestion. We also surprised with the results we got. However, it is consistent with the results about DGATs function in mammals. As we know, DGAT1 and DGAT2, are the main enzymes responsible for TAG synthesis. And they play specific roles in different tissues and in some species in TAG synthesis, especially on substrate specificity. There is plenty of evidence in mammal researches that DGAT1 and DGAT2 mediate distinct hepatic functions. DGAT2 is primarily responsible for incorporating endogenously synthesized Fatty acids (FAs) into TG, whereas DGAT1 plays a greater role in esterifying exogenous FAs to glycerol.

In addition, teleost fish utilize glucose poorly and the capacity of the pathway to convert glucose into cellular lipids for storage is relatively low (5,6). Fish meal and fish oil are the most popular energy resource for the fish. It makes sense that DGAT1 and DGAT2 are active and DGAT1 has stronger effect in fish liver which utilize lots of exogenous FAs. It is consistent that DGAT1 might be very important for hepatic TG synthesis in fish. DGAT2 also has high expression in fish liver, but de novo lipogenesis activity in liver is lower. In our previous publication, we also found DGAT1 inhibition can totally blocked triglyceride accumulation induced by oleic acid in croaker hepatocytes (7). In the end, we improved our discussion in the revised manuscript to make it better.

- 5) Viegas I, Jarak I, Rito J, et al. Effects of dietary carbohydrate on hepatic de novo lipogenesis in European seabass (*Dicentrarchus labrax* L.) [J]. *Journal of lipid research*, 2016, 57(7): 1264-1272.
- 6) Bou M, Todorčević M, Torgersen J, et al. De novo lipogenesis in Atlantic salmon adipocytes[J]. *Biochimica et Biophysica Acta (BBA)-General Subjects*, 2016, 1860(1): 86-96.
- 7) Xiang X, Han S, Xu D, et al. Effects of DGAT1 inhibition on hepatic lipid deposition, antioxidant capacity and inflammatory response in *Larimichthys crocea*[J]. *Aquaculture*, 2021, 543: 736967.

5. Absent throughout the results is a densitometric analysis of protein abundance for the Western blot assays. It is critical for protein abundance to be quantified and statistically analyzed for differential expression. Currently, evaluation of relative protein abundance can be only a subjective evaluation of how dark/thick the bands are. Given that a considerable number of inferences are made from these data, the findings generated from Western blot analyses must be supported by the appropriate analyses. Similarly, a quantitative analysis of lipid droplet number and/size is essential for images in Fig 4e.

Response: We thank the reviewer for bringing up this important point. Following the suggestion from the reviewer, we provided relative protein abundance analyses in Fig 5 and Fig 8. At the same time, we simplified the results and corrected the legends in the revised manuscript. As for lipid droplet, we tested the lipid droplet formation in specific yeast H1246 cells. The H1246 cells is a triglyceride synthesis-deficit mutant yeast and have no lipid droplet. The goal for using this line is to check whether croaker or trout DGATs contribute to lipid droplet formation. The contribution of DGATs on lipid droplet formation could be our next scientific question. And there is the limit for quantifying the lipid droplet in yeast. We will check this problem if we get the DGATs-KO cell lines.

6. Why are t-tests used in some graphs while multiple means comparisons are used other times? For example, Fig 4a uses a multiple means comparison while Figs 4b-d, despite showing the same type of analysis/data, use a t-test to compare to the control. Please be consistent. In cases where comparing between all treatment groups is valuable (such as Fig 4 or Fig 6a,b), then a multiple means comparison will be most insightful regarding relative contribution of each protein to total activity.

Response: We appreciate the reviewer's comment. Sorry for the author's carelessness. We agreed with reviewer's advice and revised the Fig4 using multiple means comparison. In Fig 4a-d, we want to show relative contribution of each protein to total activity which can help us to know the function of croaker and trout DGATs. And we use the t-test for Fig 6a,b which can show the effect on DGAT1 promoter by each transcriptional factor.

7. There should be some statistical support provided to support the statement in line 203 about the dose dependent increase in promoter activity. A simple linear regression will suffice.

Response: Thanks for the kind suggestion. Based on the results in Fig7B, it implied that the croaker DGAT1 promoter activity increased with the increasing of CREBH concentration. It would support strongly that CREBH upregulate DGAT1 promoter activity in croaker. We improved the sentence

in the revised manuscript (Line 192-193).

8. Acetate acid should be written as acetic acid.

Response: Thanks for the suggestion. The word has been revised in the revised manuscript.

9. No where in the manuscript is it indicated what inhibitors of DGAT1 and DGAT2 were used.

Response: Thank you for your kind suggestion. Sorry for the author's carelessness. According to the reviewer's advice, we supplemented the information of DGAT1 and DGAT2 inhibitor in the revised manuscript (Line 387-388).

10. Please improve clarification in the figure legends regarding what two treatments are being compared for t-tests (ie: what treatment is used for the control). In some figures like Fig 6a,b, it is not always obvious.

Response: Thanks for the kind suggestion. We improved the figure legends in revised manuscript. In Fig 6a,b, The negative control was the cells transfected with an empty plasmid with no promoter sequence (pGL6-basic) and empty PCS2 plasmid. The value was used to normalize (n=4).

11. Fig 7: Does CREBH-F mean the full-length transcript, compared to CREBH-N which is the N-terminus of the protein? Some panels in Fig 7 have CREBH while others have CREBH-F. Are they the same thing? Please clarify.

Response: Thanks for the suggestion. Yes. CREBH-F is the full-length transcript. The N-terminal fragment of CREBH (CREBH-N) (amino acids 1-291) was amplified from the full length of CREBH (CREBH-F). We clarified the abbreviation and marked it clear in the Fig 7 in the revised manuscript.

REVIEWERS' COMMENTS:

Reviewer #1 (Remarks to the Author):

I am satisfied with the revision and have no comments.

Reviewer #2 (Remarks to the Author):

The manuscript is significantly improved compared to the initial submission.

Dear Reviewers,

We thank you for your feedback and comments on our manuscript.

Reviewer 1:

I am satisfied with the revision and have no comments.

Response: We thank reviewers for their interest and enthusiasm of our work.

Reviewer 2:

The manuscript is significantly improved compared to the initial submission.

Response: We thank reviewers for his/her critical and helpful evaluation of our work.